# Live imaging of the *Cryptosporidium parvum* life cycle reveals direct development of male and female gametes from type I meronts

**Elizabeth D. English, Amandine Guérin, Jayesh Tandel, Boris Striepen** *

Department of Pathobiology, School of Veterinary Medicine, University of Pennsylvania, Philadelphia, Pennsylvania, United States of America

* striepen@upenn.edu

**Data Availability Statement:** The numerical data underlying image analysis and quantification shown in Figs 1, 2, 3, 4, 5 and S2 are available in

## Abstract

*Cryptosporidium* is a leading infectious cause of diarrhea around the world associated with waterborne outbreaks, community spread, or zoonotic transmission. The parasite has significant impact on early childhood mortality, and infection is both a consequence and cause of malnutrition and stunting. There is currently no vaccine, and treatment options are very limited. *Cryptosporidium* is a member of the Apicomplexa, and, as typical for this, protist phylum relies on asexual and sexual reproduction. In contrast to other Apicomplexa, including the malaria parasite *Plasmodium*, the entire *Cryptosporidium* life cycle unfolds in a single host in less than 3 days. Here, we establish a model to image life cycle progression in living cells and observe, track, and compare nuclear division of asexual and sexual stage parasites. We establish the length and sequence of the cell cycles of all stages and map the developmental fate of parasites across multiple rounds of invasion and egress. We propose that the parasite executes an intrinsic program of 3 generations of asexual replication, followed by a single generation of sexual stages that is independent of environmental stimuli. We find no evidence for a morphologically distinct intermediate stage (the tetraploid type II meront) but demonstrate direct development of gametes from 8N type I meronts. The progeny of each meront is collectively committed to either asexual or sexual fate, but, importantly, meronts committed to sexual fate give rise to both males and females. We define a *Cryptosporidium* life cycle matching Tyzzer's original description and inconsistent with the coccidian life cycle now shown in many textbooks.

## Introduction

Pathogen-associated diarrheal disease is one of the leading causes of mortality in children under the age of 5 years [1]. While efforts to improve sanitation, hygiene, and access to clean water have reduced the number of diarrheal deaths, nearly half a million children under the age of 5 died due to diarrheal diseases in 2015. The 3 most common causes of pathogen-associated diarrhea in children under 5 are rotavirus, *Cryptosporidium*, and *Shigella*, which together account for more than half of all diarrheal deaths in this age group [1,2]. Malnourished

the supplementary information as Excel spreadsheets S1, S2, S3, S4, and S5 Data.

**Funding:** This work was supported in part by grant R01AI127798 from the National Institutes of Health to BS and a postdoctoral fellowship from EMBO to AG (ALTF 58-2018). The funders had no role in study design, data collection and analysis, decision to publish, or preparation of the manuscript.

**Competing interests:** The authors have declared that no competing interests exist.

**Abbreviations:** BSA, bovine serum albumin; CDPK, calcium-dependent protein kinase; FBS, fetal bovine serum; GDV1, gametocyte development 1; HP1, heterochromatin protein 1; LDH, lactate dehydrogenase.

children are particularly susceptible to severe cryptosporidiosis [2–4], and, in turn, infection with this pathogen can have long-lasting consequences for the nutritional status and overall growth and development of children [5,6]. Children acquire nonsterile immunity to *Cryptosporidium* that protects from severe disease and malnutrition [7]; however, this immunity is slow to develop, and, currently, no vaccines are available to prevent the infection [3]. Nitazoxanide has been approved by the United States Food and Drug Administration for the treatment of cryptosporidiosis, but this drug is not effective in immunocompromised or malnourished individuals [8]. The last 5 years have seen a significant push toward better treatments for cryptosporidiosis (see [9] for a succinct review). Several of these efforts took advantage of recent advances in the development of antimalarials by using cherry-picked compound collections initially derived in phenotypic screens against *Plasmodium falciparum* [10–15]. The value of targeting multiple life cycle stages is a clear lesson that emerged from the malaria drug development effort [16]. A more comprehensive understanding of the biology of the *Cryptosporidium* life cycle and the relative susceptibility of its different segments is required to discover and improve drugs to establish effective treatments for this disease [3].

*Cryptosporidium* is a single cell protist parasite and a member of the phylum Apicomplexa, organisms that undergo complex life cycles of asexual and sexual reproduction. In the well-studied apicomplexans *Plasmodium* and *Toxoplasma* (the causative agents of malaria and toxoplasmosis), this life cycle unfolds in different hosts: mice and cats or humans and mosquitoes, respectively. In contrast, *Cryptosporidium* replicates both asexually and sexually within a single host and transmission between hosts occurs through meiotic spores called oocysts. In humans and animals, *Cryptosporidium* infects the intestinal epithelium and oocyst shedding begins on the third day of infection. In the absence of cell-mediated immunity, the infection is chronic and parasite growth continues unabated [17–19]. Here, we study *Cryptosporidium parvum*, a parasite of cattle and humans that is experimentally tractable [20]. At any given time, roughly one-third of the parasites within the small intestine of an infected mouse replicate asexually, one-third appear to be sexual stages, and one-third represent postfertilization stages that are in the process of forming oocysts [21]. In cell culture systems, including the widely used human adeno carcinoma cell line HCT-8, *Cryptosporidium* is limited to approximately 3 days of growth. Inoculation of HCT-8 cells with oocysts or sporozoites released from oocysts produces robust infection with parasites that reproduce asexually; however, after 2 days, the culture abruptly shifts to the sexual phase of the life cycle and is dominated by male and female gametes and growth ceases. Postfertilization stages are not observed in culture, likely due to a block in the fertilization step [21]. Interestingly, in organoid-based cultures, longer-term growth has been observed as has fertilization and oocyst formation [22–24]. Sex thus appears to reset the life cycle and initiate subsequent rounds of asexual growth and expansion.

Different apicomplexans have evolved diverging mechanisms to accommodate the progression of developmental stages to their respective host niches. Commitment to sexual development in *Plasmodium* occurs at varying frequencies depending on species and strain, suggesting an underlying inherited developmental threshold [25]. A small portion of each asexual generation commits to sexual development and initiates gametocyte development; these cells will mature into gametes once ingested by mosquitoes with a blood meal and then undergo fertilization [26]. Tissue and biochemical cues have been identified that impact on the likelihood of the developmental switch that results in exit from the asexual cell cycle and differentiation into the growth arrested gametocyte stage [27,28]. In contrast, conversion of *Hammondia* from fast-growing tachyzoites to slow-growing bradyzoites appears to be governed by a molecular clock [29,30]. Similarly, *Eimeria* executes a predetermined number of asexual cycles prior to the emergence of gametes [31,32].

The *Cryptosporidium* life cycle is remarkably short, and a substantial portion of it can be studied using tissue culture. Here, we establish a long-term live-cell microscopy model to directly observe the life cycle and to fate map developmental progression. We find no evidence of environmental induction of gametogenesis, but strict adherence to a timed developmental program. The intracellular development of all stages unfolds in roughly 12-hour intervals, with 3 generations of asexual meronts followed by a single generation of gametes. Merozoites emerging from one parasite cell are collectively committed to either an asexual or sexual fate, but sexually committed meronts give rise to both males and females. We rigorously demonstrate that gametes develop directly from asexual stages that produce 8 merozoites, known as type I meronts, and we refute a role for a morphologically distinct type II meront as an intermediate stage between the asexual and the sexual phase of *C. parvum* development.

## Results

### Sexual differentiation of *Cryptosporidium parvum* follows a parasite intrinsic program

*C. parvum* differentiates from the asexual to sexual phase of its life cycle 48 hours into culture and parasites cease to replicate. We wondered how this transition may be triggered and considered the presence of a parasite extrinsic stimulus (Fig 1A). This might include changes in the physicochemical properties of the environment [33], the depletion or accumulation of a metabolite [27,34], or the activity of a dedicated density-dependent quorum sensing mechanism [35]. Alternatively, *C. parvum* may follow an intrinsic program that is independent of extracellular factors. To test for differentiation stimuli, we performed experiments with conditioned media (Fig 1B). Media were conditioned by growing HCT-8 cells with or without *C. parvum* infection for 48 hours, the time point when differentiation occurs. The media was filtered (0.45 μm) to remove extracellular parasites and then transferred to fresh cultures. These cultures were infected with a *C. parvum* reporter line expressing nanoluciferase, and we monitored parasite growth over 72 hours by luciferase assay [20]. Use of infection conditioned media did not result in an earlier arrest of parasite growth (Fig 1C).

We also conducted experiments using a *C. parvum* reporter strain expressing histone H2BmNeon and recorded the developmental progression through different stages based on the number and morphology of their nuclei [21]. We again used conditioned media, this time added to infected coverslip cultures, which were processed for immunofluorescence assays 24, 48, and 72 hours after infection. The number of male gamonts (blue), female gametes (pink), and asexual meronts (green) was scored at each time point by microscopy (*n* = 3) and is displayed as a fraction of all parasites encountered (Fig 1D). Conditioning did not hasten life cycle progression, and the representation of different stages was indistinguishable between conditioned and unconditioned media and similar to that previously reported [21].

We considered that a sex-inducing factor might be unstable, poorly soluble, or remain cell associated and thus is not transmitted well by media transfer. We used super-infection of the same culture to test this (Fig 1E). Host cells were first infected with unmarked wild-type *C. parvum*; 24 hours later, they were infected again, this second time with a transgenic parasite strain expressing a fluorescent reporter. We then performed immunofluorescence assays to score stages and assessed life cycle progression of both infections separately. The second infection again produced gametes only after 48 hours (Fig 1F). Despite the early presence of gametes from the first infection, both waves showed similar kinetics but were offset by their 24 hours difference of time in culture. Taken together, we did not find evidence for an external induction mechanism and thus propose that *C. parvum* is following an intrinsic program of life cycle progression.

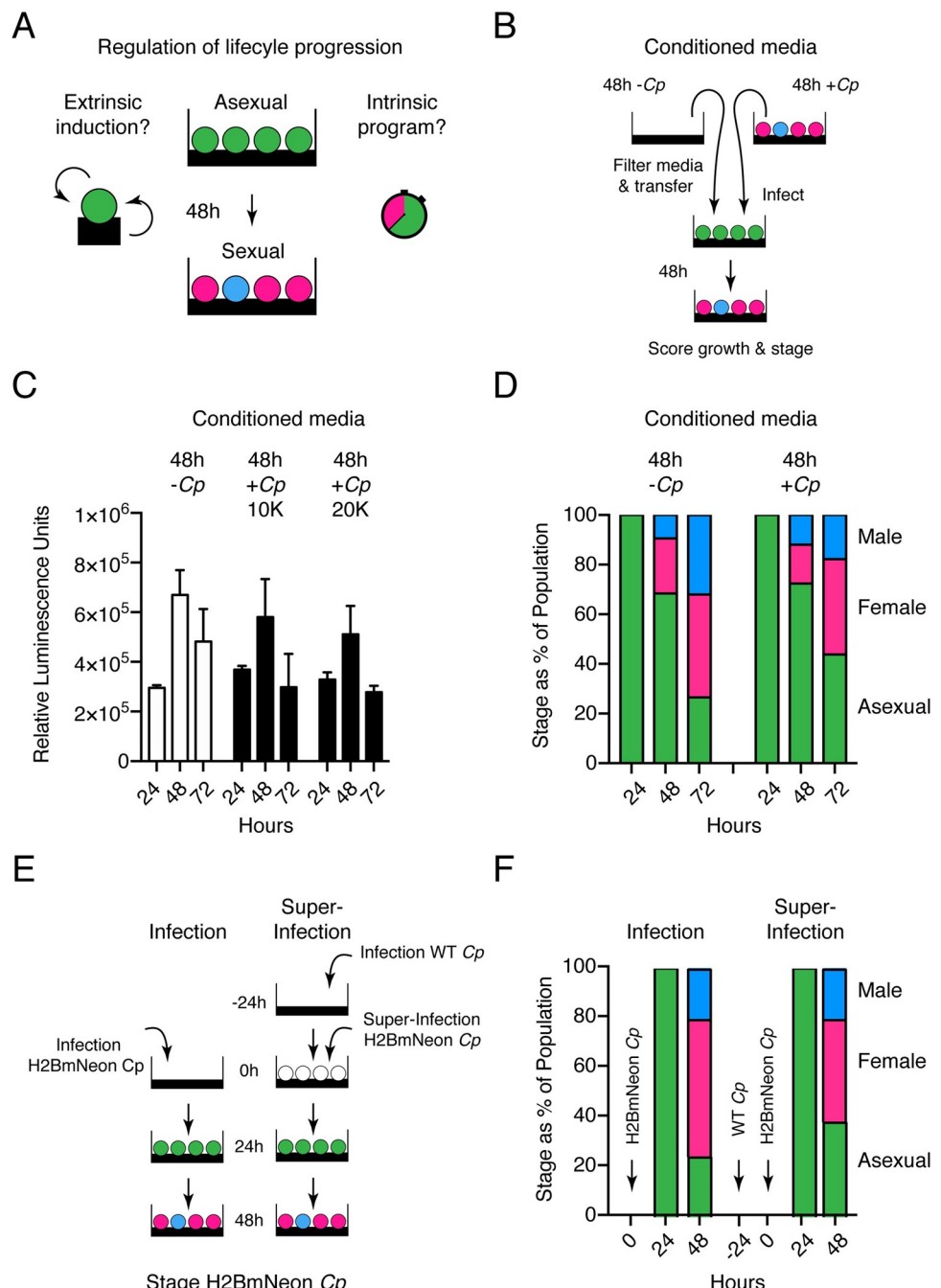

**Fig 1. Sexual differentiation of *C. parvum* is not dependent upon a secreted factor. (A)** Schematic representation of both the induction and program hypotheses of *C. parvum* sexual differentiation. **(B)** Schematic representation of the conditioned media experiment. Briefly, media was conditioned for 48 hours in the presence of HCT-8 host cells with or without *C. parvum* infection. Conditioned media was passed through a 0.45-μm filter and then used for new infections that were scored for growth **(C)** and sexual differentiation **(D)**. (C) Growth assay by luminescence for *C. parvum* growth in conditioned media. (D) Stage scoring of *C. parvum* at 24, 48, and 72 hours postinfection when grown in conditioned media. **(E)** Schematic representation of super-infection experiment. Briefly, HCT-8 cells were infected with H2BmNeon parasites that were previously infected for 24 hours with WT parasites or not previously infected. Both the primary infection (WT) and super-infection (H2BmNeon) were scored for parasite life stage at indicated time points postinfection. **(F)** Stage scoring and life cycle progression of *C. parvum* super-infection experiment. Underlying data are provided in the Supporting information as S1 Data. WT, wild type.

## Imaging the intracellular development of asexual and sexual stages of *C. parvum*

Next, we wanted to observe the *C. parvum* life cycle program in real time by live-cell microscopy. We engineered parasites to express an mScarlet fluorescent protein in the cytoplasm in addition to a mNeon-tagged H2B histone labeling the nuclei in green (S1 Fig). These parasites were used to infect HCT-8 cells grown in 8-well chamber slides and imaged using a DeltaVision OMX Structured Illumination Microscope controlling temperature and atmosphere (GE, now Leica Systems, Buffalo Groove, IL see Materials and methods for detail). In preliminary experimentation, we established that imaging every 30 minutes permitted continuous recording for up to 42 hours while maintaining parasite and host cell viability. We began imaging cultures at 11 hours postinfection, which we had previously established as the end of the first sporozoite-initiated merogony cycle [36], and we also conducted experiments imaging from 29 or 40 hours postinfection onward. We used multiple point visiting and autofocus routines to allow us to observe cells in parallel and in significant numbers, collecting a total of 6,171 hours of time-lapse data, of which 4,542.5 hours were suitable for analysis. We were able to consistently distinguish the intracellular development of asexual meronts, from that of male gamonts, and female gametes. Asexual parasites were observed prior to 40 hours postinfection, sexual stages after that point.

Fig 2A shows selected still images from 2 representative movies of asexual development (see S1 Movie for multiple additional cells). We analyzed the intracellular development of a total of 380 meronts and found a mean time of 12.57 hours from invasion to egress, which is similar to the timing of sporozoite-initiated *C. parvum* merogony [36]. *C. parvum* replicates by schizogony [37], resulting in a cell with 8 nuclei. We found that the increase in the number of nuclei strictly followed geometric progression, indicating that in contrast to *Plasmodium* [38], in *C. parvum*, nuclear divisions are highly coordinated and synchronous throughout. Using nuclear divisions as landmarks, we discerned 3 phases: a long initial establishment phase, a relatively brief mitotic phase, and a final budding phase. Meronts completed the first nuclear division after 7.92 hours and then ran through 2 additional complete mitotic cycles, taking about 1 hour for each (Fig 2C–2E). There was a lag phase of 2.77 hours between the last division and egress.

The development of male gamonts followed a pattern similar to that of meronts (Fig 2B, S2 Movie). After a 6.28 hours establishment phase, 4 rapid mitotic cycles produced 16 nuclei, and the average time to egress was 12.08 hours (Fig 2F–2H). Up to the 8N stage, the nuclei of male gamonts were round; only the last division produced the highly condensed spindle shaped nuclei, which are characteristic for male gametes [39,40]. As seen for meronts, the time required increased slightly from division to division (Fig 2F). While immature male gamonts appeared overall similar to meronts, they can be distinguished. The nuclei of gamonts clustered to the center, while nuclei of meronts showed greater dispersion (see S2 Fig). We note that half of the male parasites observed failed to egress (93 out of 172 times); this may be typical for male gamonts, associated with the culture model, or could reflect additive photodamage due to longer overall imaging times.

Female parasites did not undergo nuclear division and remained intracellular, allowing us to image them for 24 hours (the time the experiment ended). However, the size of their nuclei increased 3 to 4 times in area, and the overall size of the gamete grew 6 to 8 times in this time (Fig 2I and 2J, S3 Movie). We made 2 additional observations. First, while female gametes grew rapidly initially, growth plateaued after 12 to 15 hours. Female gametes are the transcriptionally and translationally most active of all life cycle stages as they produce essentially all components of the oocyst [21,41], and this was evident in their production of fluorescent protein. We observed a robust increase in cytosolic mScarlet fluorescence beginning at 8 hours

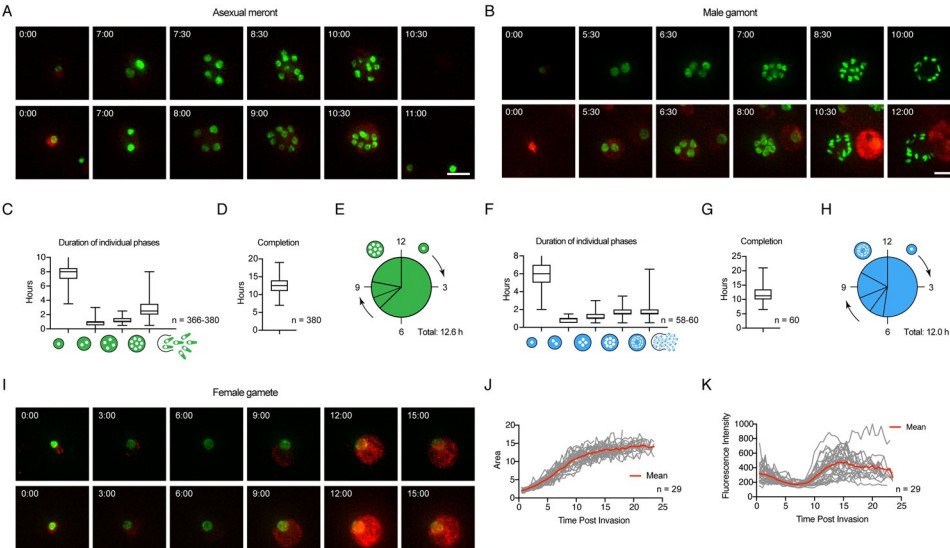

**Fig 2. The intracellular development of all stages can be observed in real time in living cells.** (A) Images taken from time-lapse microscopy depicting asexual growth and nuclear division, 2 representative meronts. Scale bar: 5 μm; insets represent hours from time of invasion. (B) Images taken from time-lapse microscopy depicting male parasite growth and nuclear division, 2 representative gamonts. Scale bar: 5 μm; insets represent hours from time of invasion. (C) Duration of phases of asexual growth. Parasites remain as a single nucleus for an average of 7.92 hours, followed by 0.83 hours as 2 nuclei, 1.11 hours as 4 nuclei, and 2.77 hours as 8 nuclei prior to egress ($n$ = 366 to 380). (D) Asexual parasites remain intracellular (invasion to egress) for an average of 12.57 hours ($n$ = 3 80). (E) Graphical representation of the timing of nuclear divisions during asexual development. (F) Duration of phases of male growth. Parasites remain as a single nucleus for an average of 6.28 hours, followed by 0.92 hours as 2 nuclei, 1.07 hours as 4 nuclei, and 1.69 hours as 8 nuclei, and 2.05 hours as 16 nuclei prior to egress ($n$ = 58 to 60). (G) Male parasites remain intracellular (invasion to egress) for an average of 12.08 hours ($n$ = 60). (H) Graphical representation of the timing of nuclear divisions during male development. (I) Images taken from time-lapse microscopy depicting female growth, 2 representative gametes. Scale bar: 5 μm; insets represent hours from time of invasion. (J) Total area of female parasites over time. Gray lines represent individual female parasites, and red line indicates the average area of female parasites over time ($n$ = 29). (K) Total fluorescent intensity of cytoplasmic mScarlet for the entire area of female parasites over time. Gray lines represent individual female parasites, and red line indicates the average of female parasite over time ($n$ = 29). Underlying data are provided in the Supporting information as S2 Data.

(Fig 2K); this fluorescence reached peak intensity 12 to 14 hours after invasion (note that we did not correct for photobleaching and that fluorescence thus diminishes in the absence of new synthesis). We note that both these time frames match the time to egress for male gametes. Interestingly, all 3 parasite stages are morphologically indistinguishable for the first 6 to 8 hours after invasion, after which male and asexual parasites begin to divide their nuclei, while female parasites continue to increase in size (S3 Fig).

## Long-term imaging reveals 3 generations of meronts followed by gametes

Our imaging covered a total of 60 hours of parasite development in multiple overlapping experiments. We tracked hundreds of individual parasites and mapped them onto the overall life cycle timeline. Fig 3A shows each cell as a line, with the start representing invasion, and the end showing the time of egress (egress is only shown for meronts (green) and male gametes (blue) as female parasites do not egress). We next mapped all observed nuclear divisions, as well as invasion and egress events as individual time points (Fig 3B–3E). These analyses revealed 3 waves of merogony followed by a single wave of gamete development (please note that we only observed the tail end of the first wave as we start imaging at 11 hours). Waves begin with a relatively sharp line followed by a trail of "late comers." We note a consistent shift

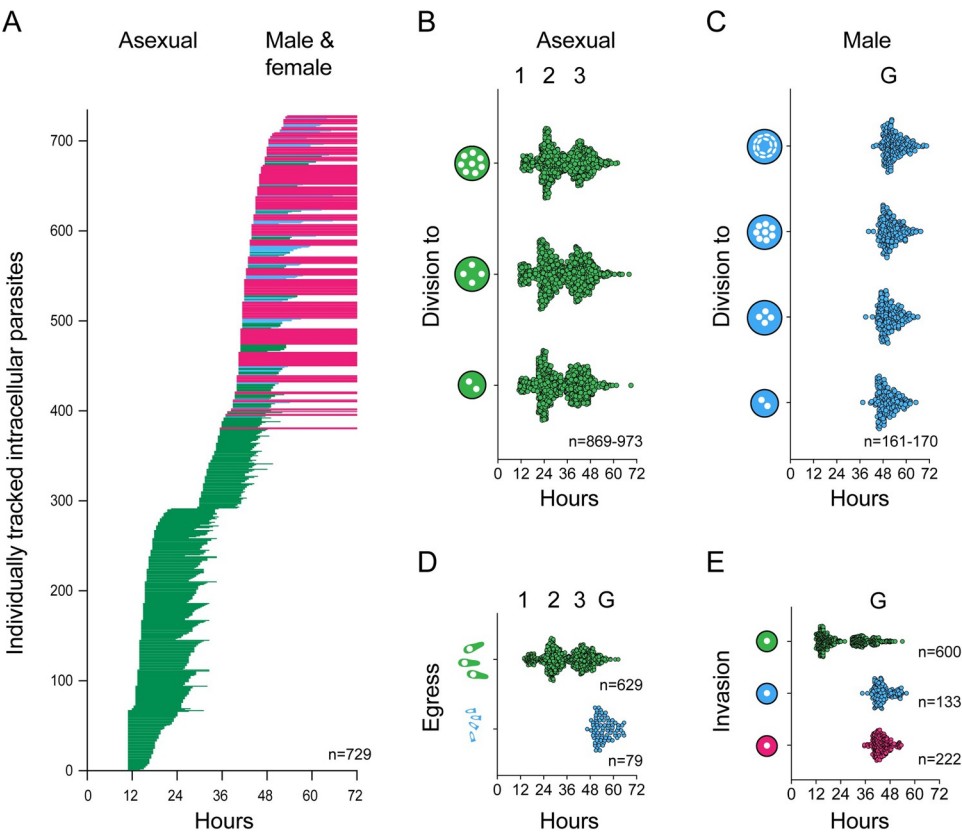

**Fig 3. Extended time-lapse imaging reveals 3 cycles of asexual schizogony followed by differentiation into gametes. (A)** Individually tracked parasites over time. Each horizontal line represents an individual parasite from invasion to egress (green = asexual, blue = male, and pink = female). Lines for females cutoff at 72 hours, as we do not observe egress of these parasites. Parasites with observed egress, but which invaded prior to the start of the experiment are included with lines beginning at 11 hours (*n* = 729). **(B)** Individually plotted nuclear division events for asexual parasites over time. The time point at which we observe 2, 4, or 8 nuclei for the first time for parasites designated as asexual are plotted as individual points over time (*n* = 869 to 973). **(C)** Individually plotted nuclear division events for male parasites over time. The time point at which we observe 2, 4, 8, or 16 nuclei for the first time for parasites designated as male are plotted as individual points over time (*n* = 161 to 170). **(D)** Individually plotted egress events for both asexual (green) and male (blue) parasites over time. The time at which we no longer observe a parasite is designated the time of egress. Each point represents a single asexual meront or male gamont. There are 3 distinct clusters of asexual egress events and a single cluster of male egress events (asexual *n* = 629, male *n* = 79). **(E)** Individually plotted invasion events for asexual (green), male (blue), and female (pink) parasites over time. The time at which a parasite is first observed is considered the time of invasion. There are 2 observed clusters of invasion events leading to asexual parasites and only 1 cluster leading to male or female cells, respectively (asexual *n* = 600, male *n* = 133, and female *n* = 222). Underlying data are provided in the Supporting information as S2 Data.

to gametes at 40 hours in line with previous studies using immunofluorescence assays [21,42,43].

## Meronts release merozoites committed to either asexual or sexual fate

Next, we searched our image dataset for instances in which we could track successive generations of parasite development. Fig 4A shows selected frames over 27 hours of a time-lapse movie (S4 Movie). The fate of mature cells is indicated by a colored arrowhead, and newly invaded next generation stages are highlighted by white arrowheads. We used this information to derive trees mapping the fate of each parasite cell (Fig 4B shows examples with the interpretation of the cells shown in Fig 4A in the middle). Note that we cannot unambiguously map

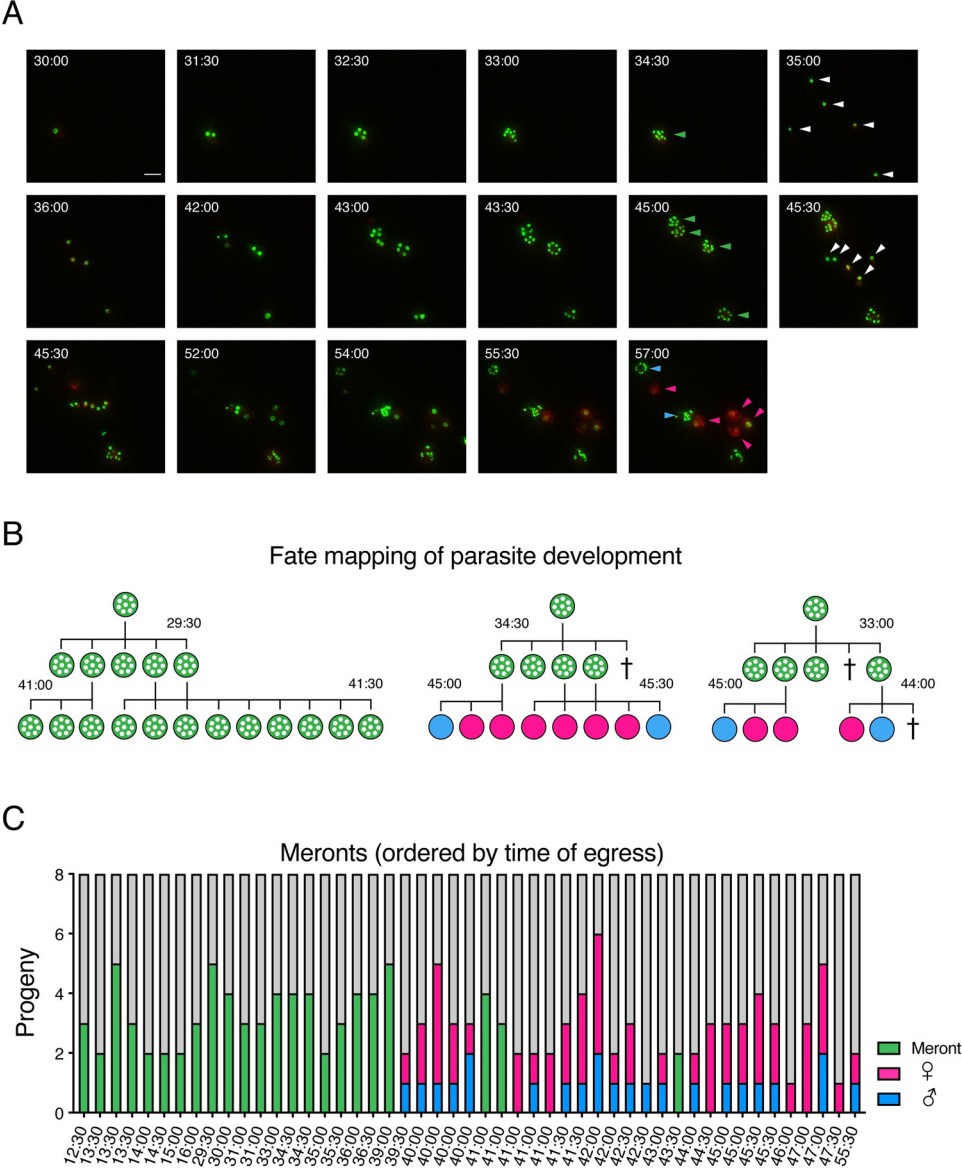

**Fig 4. Meronts commit their merozoites to either asexual or sexual fate, but when committed to sex give rise to both males and females. (A)** Images taken from time-lapse microscopy depicting multiple generations of parasite replication, including both asexual and sexual stage growth and development. Stills take from video (see S4 Movie) over 27 hours of observation; insets represent hours from time of infection of the culture. Green arrows indicate final fate of asexual meronts, and white arrows indicate newly invaded parasites immediately following an egress event. Blue arrows indicate parasites designated with a male fate, and pink arrows indicate parasites designated with a female fate. Scale bar: 5 μm. **(B)** Representative trees depicting the mapped fate of parasites for which the fate of progeny of a single parental meront could be tracked. The tree in the center represents the images seen in (A). Times are shown in hours postinfection. Crosses indicate cells that were lost or died before fate could be assigned. **(C)** Fate of progeny of single 8N meronts, ordered by time of egress. For each parental meront, the fate of all observable progeny was tracked and is indicated (asexual = green, male = blue, female = pink, and lost to observation or ambiguous = gray). Individual parental meronts are ordered by the time at which the parental meront was last observed. Underlying data are provided in the Supporting information as S3 Data.

the origin or fate of all parasites. Merozoites left the field of view, cells were lost due to egress and/or cell death, the simultaneous egress of multiple meronts obscured origin, or phototoxicity stalled the development of some parasites before we could assign fate. Nonetheless, we were

able to track the fate of the progeny of 49 meronts. During earlier time points, all merozoites emerging from a meront again gave rise to asexual meronts in the next generation (Fig 4C). As the cultures shifted from their asexual to their sexual phase at 40 hours, we found all progeny of individual meronts to give rise to sexual stages. Importantly, when tracking the offspring of individual meronts, we never observed both asexual and sexual stages developing from the same parental meront. We conclude that merozoites are collectively committed to life cycle transition, which suggests that commitment is most likely to occur during the intracellular development of the meront.

## Meronts committed to sexual development give rise to both male and female gametes

While merozoites from a single meront were strictly committed to either asexual or sexual fate, they were not collectively committed to a single specific sex. We observed both male and female offspring from the same meront (Fig 4C). Out of 26 individual meronts committed to sexual fate for which we were able to observe the development of at least 2 offspring, we observed both male and female offspring 19 times. We were never able to observe all 8 progeny from any meront; the maximum number of observed offspring was 6, in which 2 developed into males and 4 became female. To determine the male to female ratio of offspring from a single meront, we conducted a weighted confidence interval based on all 73 observed offspring from the 26 sexually committed meronts. Based on our observations, the 95% confidence interval suggested that out of the 8 progeny in each meront, between 1.92 and 3.11 will be male.

## Gametes develop directly from type I meronts; type II meronts are not apparent

Currently, many life cycles of *Cryptosporidium* depict a morphologically distinct tetraploid generation of asexual parasites called the type II meront, as an intermediate between asexual meronts and gametes (see, e.g., the widely reproduced life cycle from the Centers of Disease Control and Prevention at https://www.cdc.gov/parasites/crypto/pathogen.html). Type II meronts are shown to give rise to 4 merozoites in contrast to asexual type I meronts, which produce 8. This model predicts sexual differentiation to be preceded by a wave of tetraploid meronts (Fig 5A); surprisingly, we did not observe this in a previous study that used molecular markers to define stages [21]. Our live-cell imaging experiments used a nuclear marker that clearly distinguished 4 and 8 nuclei stages and thus provided the opportunity to test this rigorously using a large dataset. We analyzed the eventual fate of 1,095 parasites that reached the 4N stage across the 60 hours observed. Parasites were binned by the time at which 4 nuclei were observed and then categorized into 1 of 3 outcomes (Fig 5B): (1) disappearance after the 4N stage consistent with egress predicted by type II merogony; (2) progression to 8N prior to egress (predicted by type I merogony); or (3) progression to 16N prior to egress (male gamogony). Parasites that remained 4N until the end of the imaging experiment were excluded from analysis. From 11 to 40 hours postinfection, the vast majority of parasites that reach the 4N stage continued past that stage to the 8N stage prior to egress (Fig 5C). During this time, we did not observe male (16N) parasites. After 40.5 hours in culture, the proportion of the population that egresses at the 8N stage decreased markedly over time, while the proportion of the population that develop into males increased at the same rate. Parasites with apparent egress at the 4N stage were rare, and, importantly, their frequency did not change over the culture time and life cycle.

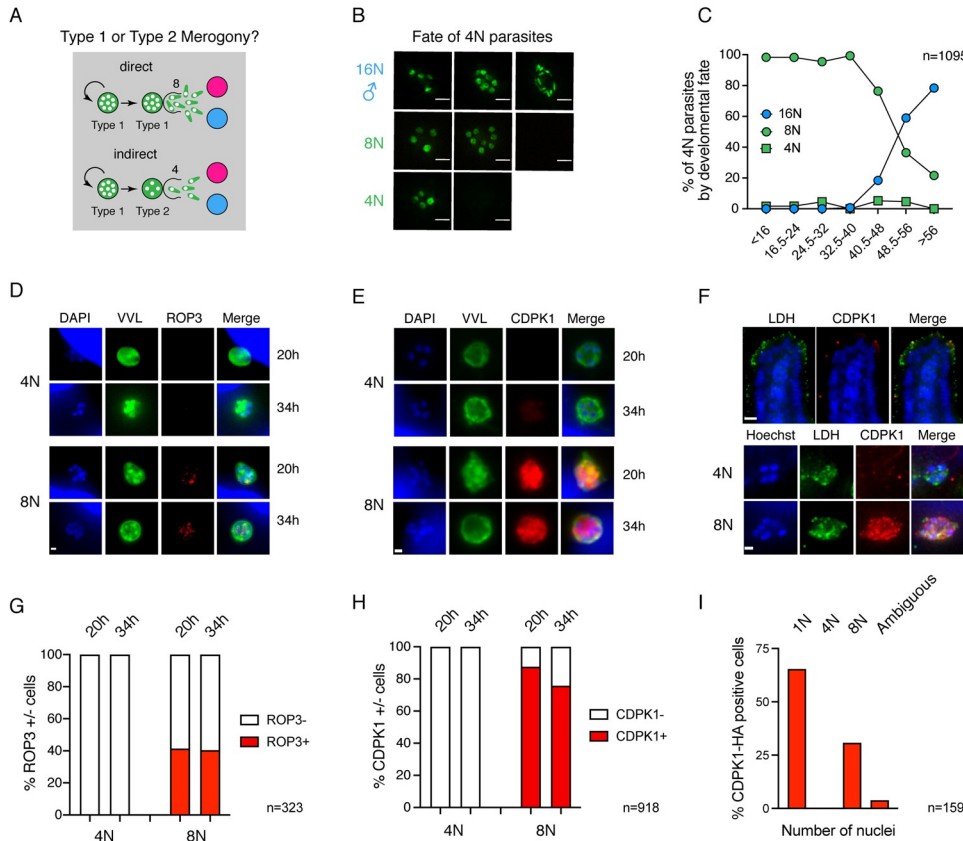

**Fig 5. Development to gametes occurs directly from type I meronts, and there is no evidence to support a 4N type II meront as the committed stage. (A)** Schematic depicting hypotheses of life cycle progression. Parasites might undergo direct development in which type I meronts producing 8 progeny develop into sexual stages or indirect development in which an intermediary type II meront producing only 4 progeny gives rise to sexual stages. **(B)** Images depicting the possible fates of parasites that progress through a stage with 4 nuclei. Parasites were assigned 3 fates: (1) those that progress from 4, to 8, and eventually 16 nuclei are male; (2) those that progress from 4 to 8 nuclei are asexual type I meronts; and (3) those that appear to egress as 4 nuclei are asexual type II meronts. Parasites that remained as 4 nuclei until the end of the experiment were excluded from this analysis. **(C)** Graph of the fate of 4N parasites over time. Prior to 16 hours of infection, nearly 100% of the parasites that pass through a 4N stage become 8N prior to egress. This decrease after 40.5 hours of infection, and this decrease is proportional to the increase in parasites that become 16N (male) at these time points ($n$ = 1,095). **(D** and **E)** Immunofluorescence of HCT-8 cells infected with transgenic *C. parvum* expressing ROP3-HA or CDPK1-HA, respectively. Representative images of 4N or 8N meronts at 20 and 34 hours postinfection are shown. Only parasites with 8 nuclei express ROP3 (D) or CDPK1 (E) at either time point. Scale bar: 1 μm. **(F)** Ifnγ$^{-/-}$ mice were infected with CDPK1-HA parasites and immunofluorescence staining was conducted on frozen sections of the small intestine. Top, low magnification micrograph of a highly infected segment of the intestinal tissue. Scale bar: 15 μm. All parasite stages are labeled with an antibody to LDH (green) and mature meronts ready to egress are labeled with CDPK1 (red). Bottom, higher magnification images of 4N or 8N parasite. Parasites with 8 nuclei but not 4 nuclei express CDPK1 in vivo. Scale bar: 1 μm. **(G)** Quantification of the ROP3 positive meronts for the entire 4N or 8N population at 20 and 34 hours postinfection. **(H)** Quantification of the CDPK1 positive meronts at 20 and 34 hours postinfection. **(I)** Quantification of the number of nuclei of a total of 159 CDPK1 positive parasites observed in 26 independent fields of view of intestinal sections. Only young 1N trophozoite and meronts 8 nuclei were positive for CDPK1, and in matching our studies in culture, we did not observe a single positive tetraploid parasite. Underlying data are provided in the Supporting information as S4 Data. LDH, lactate dehydrogenase.

Loss of fluorescence, used here as a proxy for egress, is indistinguishable from host cell death or lysis due to the egress of another parasite in cells carrying multiple infections. We thus sought to independently evaluate whether 4N meronts do or do not produce merozoites and egress using molecular markers. Apicomplexan parasites assemble the organelles required

for invasion at the very end of the cell cycle, and in those parasites replicating by schizogony, rhoptries and micronemes are only apparent in formed merozoite immediately preceding egress [37,44]. Several rhoptry bulb proteins were recently described for *C. parvum*, and their expression was found to be similarly restricted to parasites harboring merozoites poised to egress [36]. We evaluated 2 time points (20 and 34 hours) to observe meronts committed to asexual or sexual fate in the next generation, respectively, and scored the expression of ROP3-HA in 4N and 8N parasites by immunofluorescence. Labeling was exclusively found in 8N parasites regardless of the point in the progression of the life cycle they were observed (Fig 5D and 5G).

Key transitions in the intracellular development of apicomplexan parasites are regulated by the activity of calcium-dependent kinases (CDPKs) [45]. CDPK1, an important drug target in *C. parvum* [46], was recently shown to underly strict cell cycle regulation. The protein was only detectable in late stages prior to egress and in very young stages following invasion [47]. We scored the presence of CDPK1-HA in 4N and 8N parasites by immunofluorescence at 20 and 34 hours. CDPK1 staining was only found in 8N stages (Fig 5E), and quantification showed this to be highly reproducible (Fig 5H). We considered that there might be differences between parasite development in vitro and in vivo and thus infected mice with CDPK1-HA parasites. Mice were killed at the peak of infection, and small intestines were resected, fixed, frozen, and sectioned (see Materials and methods for detail). Cryosections of the tissue were incubated with antibodies to HA, lactate dehydrogenase (LDH; a marker of all parasite stages), and Hoechst to label DNA. We identified a total of 159 CDPK1 positive cells in 26 fields of view for which we then scored the number of nuclei in the Hoechst channel. All positive cells had a single or 8 nuclei, and, again, we did not detect CDPK1 in 4N parasites (Fig 5F). Taken together, our culture and animal studies find no evidence for a tetraploid type II meront stage, and we conclude that development to gametes occurs directly from meronts that produce 8 merozoites. Importantly, we directly observed this transition (e.g., Fig 4A and 4B) in our time-lapse experiments 26 times.

## Discussion

Apicomplexa undergo a cascade of developmental changes as they transition through their life cycles. A complex succession of morphological types are specifically adapted to the tasks of invasion and intracellular replication in different hosts, organs, and tissues. As apicomplexans are single celled organisms, differentiation is not terminal or rigidly inherited, but rather a continuous flow, in which each generation elaborates a transient fate. Edward Tyzzer in his initial description of the *Cryptosporidium muris* life cycle identified 3 intracellular stages: microgamonts that produced 16 microgametes, macrogamonts that produced single macroga-metes, and asexual schizonts. He commented that "the number of merozoites produced in this process of schizogony is almost invariably eight" [48]; he also described fertilization and oocyst formation resulting in parasite stages containing 4 sporozoites. The concept of the tetraploid type II meront as a developmental intermediate between the asexual and sexual reproduction was introduced by John Vetterling in 1971 [49,50] after studying *Cryptosporidium wrairi* in guinea pigs. This concept might have been inspired by his extensive work on *Eimeria* in vari-ous animals where distinct meront types occur [51,52]. At the time, *Cryptosporidium* and *Eimeria* were seen as closely related members of the Coccidia (a phylogenetic view no longer held [53,54]). Vetterling's 2 meront model has been cited widely since [55] and has become the text book life cycle for *Cryptosporidium*. The core of the argument between these authors was how to interpret the tetraploid intracellular parasites found in infected animals and cultures. Are they mature meronts that will yield 4 merozoites committed to sexual differentiation, or

are these immature stages that will undergo further nuclear divisions or form oocysts? This question was difficult to resolve using fixed samples, and we therefore chose to study living cells. We documented the fate of more than a thousand tetraploid parasites by time-lapse microcopy, and our observations are entirely consistent with Tyzzer's original assertion that all meronts produce 8 merozoites—we find no evidence for a type II meront. Molecular markers that report on parasite cell cycle progression further refute type II meronts in culture and in infected animals. We note that we have not tested *C. wrairi*, the guinea pig parasite Vetterling used in his original work; however, for *C. parvum*, the most widely studied species of this parasite genus, we demonstrate a simple and direct life cycle of only 3 morphologically distinct intracellular stages: meronts that yield 8 merozoites, male gamonts, and female gametes (Fig 6A).

The intracellular development of *Cryptosporidium* appears modular. Initially, all intracellular stages, regardless of their eventual fate, are morphologically indistinguishable. Establishment of the replicative niche relies on proteins injected during and following invasion [36,56]. It takes about 5 hours to implement the export system that delivers some of these factors to the host cytosol for both asexual and sexual stages. Following the initial establishment phase, the cellular programs diverge markedly leading to asexual merozoites or male and female gametes. Intracellular development is highly synchronous and yields a stage-specific number of progeny over a roughly 12-hour time frame (8 merozoites, 16 male gametes, and a single female gamete). This is fast when compared to other apicomplexans, where each intracellular cycle unfolds over days, and may represent an adaptation to the intestinal epithelium with its high directional turnover from the crypt to the tip of each villus.

A striking feature of *C. parvum* development is the dramatic switch from asexual to sexual reproduction following 3 generations of meronts, with gametes discernable at 48 hours [21,42,43]. In our experiments, we found no evidence for an environmental sex-inducing factor. The parasite adhered to a rigid timetable of differentiation in different scenarios of media transfer or coinfection, suggesting an intrinsic developmental program. The program appears to be reset by sex, a model that is consistent with the link between sex and growth observed by multiple investigators [21–23]. This model contrasts with *Plasmodium*, where transition to sex is sensitive to environmental and metabolic indicators and stressors [27,28], which is critical to achieving balance between colonization and transmission [57]. To understand how *Cryptosporidium* might be able to forgo such regulation, it is important to consider that, in contrast to *Plasmodium*, for this parasite, sex is not solely linked to transmission but contributes to continued infection. Here, the balance between colonization and transmission is shifted to the oocyst and the likelihood that sporozoites excyst immediately. The parasite may be able to integrate environmental cues into this step. Interestingly, some authors reported thin shelled and thick shell oocysts that may morphologically reflect the dichotomy of local reinfection or transmission [40]; however, experimental confirmation of such a mechanism is lacking.

It is technically difficult to establish whether the parasite adheres to its rigid pattern or timing in animals. However, we believe this to be likely and note that oocyst shedding is detectable on the third day of infection of mice, consistent with the time frame observed in culture [18]. The mechanism underlying this intrinsic life cycle transition is unknown; the parasite may measure time, the number of intracellular cycles, or the accumulation or depletion of a particular molecule or epigenetic mark. A variety of such mechanisms have been explored in the context of the self-limiting expansion of stem cell populations [58], and this may stimulate future studies. Among Apicomplexa, the transition from asexual to sexual stages has been most intensively studied in *Plasmodium* [59], where the transcription factor AP2-G was shown to be required for gametocyte production [60,61]. AP2-G acts as a master regulator of sex-specific gene expression through a cascade of transcription factors and additional regulatory

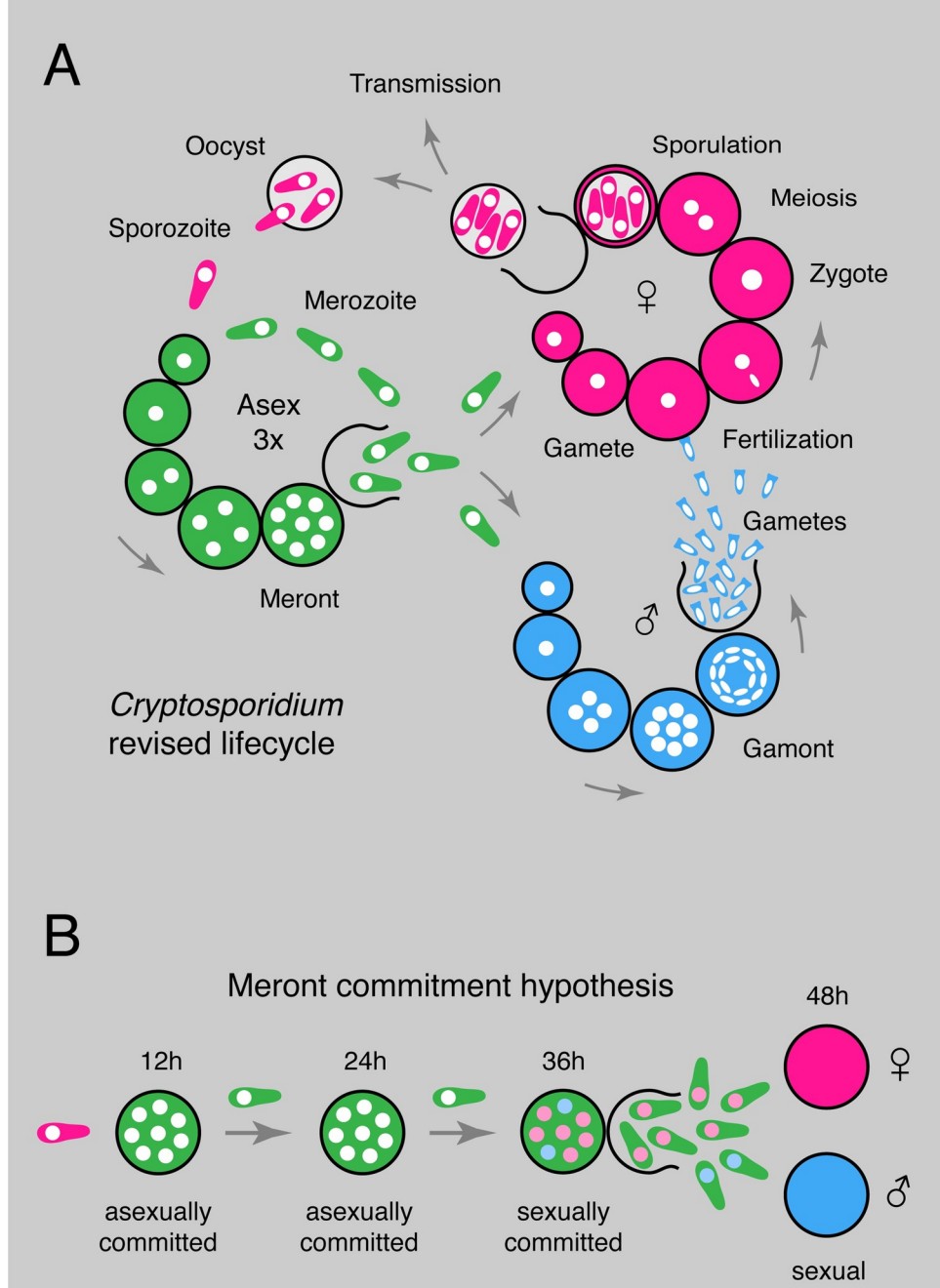

**Fig 6. Model of *Cryptosporidium* life cycle and sexual commitment. (A)** Schematic representation of the life cycle for *Cryptosporidium* summarizing the findings of this study. Infection begins with the oocyst that releases 4 sporozoites that invade intestinal epithelial cells. The parasites replicate asexually (green) by synchronous schizogony for 3 cycles and invariantly produce 8 merozoites. Merozoites emerging from the third round upon reinvasion give rise to sexual stages, both males (blue) and female (pink). The male gamont undergoes 4 rounds of synchronous nuclear division producing 16 gametes, while the female gamete is cell cycle arrested and remains haploid while expanding in size and stockpiling proteins, lipids, and carbohydrates for the future oocyst. Male gametes egress and fertilize intracellular female gametes. Following fertilization, meiosis and sporulation oocysts are released from the host cell and are immediately infectious. Oocysts can be shed with the feces resulting in transmission or excyst and reinfect the same host. **(B)** We propose a developmental commitment model of life cycle progression for *Cryptosporidium*. Merozoites emerging a merogony cycle are collectively committed to an asexual or sexual fate and, when sexually committed, give rise to both male and females (commitment is represented symbolically here by coloring the nuclei forestalling future fate; however, we note that the mechanism is unknown and may be independent of the nucleus).

genes [61]. AP2-G itself is epigenetically silenced in a heterochromatin protein 1 (HP1)-dependent manner during asexual growth [62], and HP1 silencing is removed by the protein gametocyte development 1 (GDV1) [63], which is expressed when *Plasmodium* parasites are grown under conditions that favor sexual stage development. While there are no obvious homologs of HP1 and GDV1 in *Cryptosporidium*, a similar epigenetic switch could nonetheless underlie the sexual commitment that we observed here. Very little is known about epigenetic regulation in *Cryptosporidium*, but the *C. parvum* genome encodes histone modification enzymes [64], and the parasites are susceptible to pharmacological inhibition of these pathways [65,66].

While sex is an ancient phenomenon found in most eukaryotes, how the specific sex or mating type of individuals is determined is varied and evolutionary malleable [67]. Sex can be inherited, be determined by environmental factors like temperature, or be "negotiated" between different members of a population by social behavior, biochemical clues, or cell–cell interaction and associated signaling events [68–73]. How sex is determined in Apicomplexa is unknown [31,74], but sexual differentiation and sex ratio play important roles in transmission and environmental adaptation [57]. In *Plasmodium*, the offspring of individual schizonts is thought to be collectively committed to a male or female fate in a mutually exclusive fashion [75] (see [76] for potential exceptions). One of the most interesting observations of our study in *Cryptosporidium* is that the sexually committed meront consistently gives rise to both males and females (Fig 4). We propose that the future sex of individual merozoites, and thus the sex ratio, is determined in the meront prior to egress (Fig 6B). The mechanism by which choice is achieved remains to be discovered. The fact that commitment occurs over the backdrop of cell division may offer hypotheses. Unequal inheritance or asymmetrical segregation of fate determining organelles or molecules is well established in the differentiation of mammalian cells and embryos [77,78] and could be at play here. We note an abundance of sex-specific noncoding RNAs in *Cryptosporidium* [79] as one set of potential candidates. There is also emerging understanding of mechanisms of differential inheritance of epigenetic modifications at different steps of the cell cycle [80]. The choice mechanism may well be stochastic instead of directive, as it may not matter which specific merozoite adopts a certain fate as long as the desired sex ratio is achieved. An important biological consequence of the *Cryptosporidium* commitment model is that male and female gametes by default will develop in close spatial proximity. Even at very low initial parasite burden, this mechanism will provide a safe route toward fertilization through selfing. That may be critical for a parasite that relies on a sexual life cycle reset for continued growth [21–23]. *Cryptosporidium* provides the opportunity to analyze the fundamental molecular tenants of apicomplexan life cycle progression, sex determination, and gamete interaction in a single host life cycle that we show to unfold in only 4 infectious cycles and 3 morphological stages.

## Materials and methods

### Generation of transgenic parasites

Transgenic parasites were generated using previously described methods [20,81]. Briefly, $5 \times 10^7$ *C. parvum* oocysts (Iowa II strain obtained from Bunchgrass Farms Deary, ID) were bleached and washed prior to incubation in 0.8% sodium taurocholate for 1 hour at 37˚C to induce excystation. Following excystation, a CRISPR guide plasmid and repair template with 50 base pair homology arms were introduced via nucleofection using the AMAXA 4D Nucleofector (Lonza, Basel, Switzerland). Transfected sporozoites were then diluted in PBS and used to infect a cage of ifng$^{-/-}$ mice (Jackson Laboratories Strain 002287 bred inhouse, Bar Harbor, ME). Here, we used the modified protocol for infection in which mice were given 100 μL of 8% sodium bicarbonate solution by oral gavage 10 minutes prior to oral gavage with the

transfected parasites [81]. Stable transformants were selected with 16 mg/mL paromomycin (in drinking water at libitum), and parasite shedding was monitored via nanoluciferase activity in the feces of infected mice [20].

Transgenic parasites were isolated from feces by sucrose floatation followed by cesium chloride gradient [82] and then stored at 4°C in PBS until used for each experiment.

### Immunofluorescence assay

Human adeno carcinoma HCT-8 (ATTC CCL-244) cells were maintained by serial passage in RPMI-1640 containing 10% fetal bovine serum (FBS). Immediately prior to infection, the medium was exchanged to RPMI-1640 containing 1% FBS. HCT-8 cells were infected with oocysts that were bleached using 10% bleach for 10 minutes at 4°C, washed 3 times with cold PBS, excysted with 0.8% sodium taurocholate at 37°C for 10 minutes, and washed once with PBS before addition to the culture. Infected HCT-8s were fixed using 4% paraformaldehyde for 10 minutes and then washed and permeabilized with 0.25% Triton X-100 for 10 minutes. Fixed and permeabilized cells were then blocked with 3% bovine serum albumin (BSA) for 1 hour at room temperature prior to incubation with primary antibodies for 1 hour at room temperature.

For immunohistology, ifng$^{-/-}$ mice (Jackson Laboratories Strain 002287 bred inhouse) were infected with 10,000 CDPK1-HA oocysts (a kind gift of Dr. Sumiti Vinayak, University of Illinois Urbana-Champaign), and their intestines were resected and "swiss-rolled" prior to fixation overnight in formalin. Cryosectioning was performed by the PennVet Pathology core facility, and immunofluorescence was performed as described [18].

Antibodies and dye used: anti-HA (Roche clone 3F10), VVL-FITC (Vector FL1231), and anti-LDH; a kind gift of Dr. Guan Zhu Texas A&M University, now Jilin University [83]).

Following incubation with primary antibodies, coverslips were washed 3 times with PBS at room temperature and then incubated with appropriate secondary antibodies in 3% BSA for 1 hour at room temperature. Cells were counterstained with DAPI or Hoechst for 10 minutes, washed with PBS twice and then mounted on slides with Fluoromount or Vectashield. Coverslips were observed using a Leica DM6000B Upright Widefield Microscope using 63× or 100× objectives or a GE DeltaVision OMX Structured Resolution Microscope using a 60× objective. Both microscopes are maintained by the Penn Vet Imaging Core (Philadelphia, PA).

### Acquisition and processing of live-imaging data

Eight-well chamber slides (ibidi, Gräfelfing, Germany) were seeded with HCT-8 cells, which were then grown in RPMI-1640 supplemented with 10% FBS at 37°C, 5% CO2 for 24 to 48 hours prior to infection. Host cells were switch to prewarmed RPMI-1640 supplemented with 1% FBS immediately prior to infection with bleached, washed, and excysted sporozoites. Imaging was preformed beginning at various time points using a GE DeltaVision OMX Structured Resolution Microscope using the conventional light path. Growth conditions were maintained throughout each imaging experiment at 37°C, 5% CO2, and 40% to 60% humidity. Using the AquireSR Acquisition control software, Z-stacks were taken in both the 488 and 568 channels for multiple points of interest every 30 minutes for up to 42 hours. Images were processed using the softWoRx image reconstruction and analysis software. Briefly, images were deconvolved; the channels were then aligned, and the z-stacks were compressed to generate a 2-channel image for each time point. Manual drift correction was applied in ImageJ to generate movies and stills of individual developing parasites. A total of 10 independent experiments were performed; we collected a total of 6,171 hours of images and were able to analyze 4,542.5 hours containing growth and replication information for 1,365 individual intracellular parasites.

### Analysis of live-imaging data

Analysis was performed using 2-channel time series data obtained as described above. Parasites were manually tracked and recorded for time of invasion, any subsequent nuclear replication events, and apparent egress. Data were graphed and analyzed using GraphPad Prism (San Diego, CA) and Microsoft Excel software (Seattle, WA).

### Statistical analysis

The weighted confidence interval was performed in Microsoft Excel using the observational data from 73 offspring of 26 sexually committed meronts. All other statistical analyses were performed using GraphPad Prism.

### Ethics statement

All protocols for animal experimentation were approved by the Institutional Animal Care and Use Committee of the University of Pennsylvania (protocol #806292).

### Supporting information

**S1 Fig. Generation of mScarlet-H2BmNeon parasites. (A)** Schematic overview of the guide and repair constructs used to generate the transgenic parasite line with a cytosolic mScarlet and a nuclear mNeon inserted into the TK locus. **(B)** Visualization of the fluorescent protein localization in multiple life stages for the mScarlet-H2BmNeon transgenic parasites. TK, thymidine kinase.
(TIF)

**S2 Fig. The nuclei of dividing male parasites cluster closer together than those of dividing asexual parasites. (A)** Two representative images depicting asexual parasites at 4 and 8 nuclei, including the boundary drawn around all nuclei to measure nuclear spread. The area for each meront is included. **(B)** Representative images depicting male parasites at 4, 8, and then 16 nuclei, including the boundary drawn around all nuclei at 4 and 8 nuclei stages to measure nuclear spread. The area for each gamont is included. **(C)** Comparison of the area of nuclear spread for asexual and male parasites with 4 or 8 nuclei. The area taken up by male nuclei is significantly smaller than the area taken up by asexual nuclei at both the 4 and 8 nuclei stages (Welch $t$ test, *** $p < 0.0001$). **(D)** Three representative image series of male nuclear development. Nuclei remain round when 8 nuclei are present and adopt distinct bullet-like male shape only after dividing to 16 nuclei. **(E)** Schematic representation of the development of the male gamont. Underlying data are provided in the Supporting information as S5 Data.
(TIF)

**S3 Fig. The establishment phase of asexual, male, and female parasites is visually indistinguishable for the first 6 hours of infection. (A)** Images from 2 representative asexual meronts shown every hour for the first 6 hours, followed by a 13-hour time point to confirm stage. **(B)** Images from 2 representative male gamonts shown every hour for the first 6 hours, followed by a 13-hour time point to confirm stage. **(C)** Images from 2 representative female gamonts shown every hour for the first 6 hours, followed by a 13-hour time point to confirm stage.
(TIF)

**S1 Movie. Growth and division of asexual *C. parvum*.** Five representative videos of individual asexual meronts from first appearance to apparent egress.
(MP4)

**S2 Movie. Growth and division of male *C. parvum*.** Five representative videos of individual male gamonts from first appearance to apparent egress.
(MP4)

**S3 Movie. Growth and development of female *C. parvum*.** Five representative videos of individual female gamonts from first appearance past the point of maximum fluorescence intensity.
(MP4)

**S4 Movie. Multiple generations of asexual growth and sexual development of *C. parvum*.** A representative image of 2 generations of asexual growth, followed by the development of male and female gamonts.
(MP4)

**S1 Data. Excel spreadsheet containing the underlying numerical data for Fig 1C, 1D, and 1F.**
(XLSX)

**S2 Data. Excel spreadsheet containing the underlying numerical data and summary information for all live microscopy experiments conducted for this study displayed in Figs 2C, 2D, 2E, 2F, 2G, 2H, 2J, 2K, 3A, 3B, 3C, 3D, 3E, and 5C.**
(XLSX)

**S3 Data. Excel spreadsheet containing the underlying numerical data and statistical analysis for Fig 4C.**
(XLSX)

**S4 Data. Excel spreadsheet containing the underlying numerical data for Fig 5G, 5H, and 5I.**
(XLSX)

**S5 Data. Excel spreadsheet containing the underlying numerical data for S2A, S2B, and S2C Fig.**
(XLSX)

## Acknowledgments

We are grateful to Drs. Vinayak and Zhu for sharing reagents with us and to the PennVet pathology and imaging core for assistance. We would also like to thank members of our laboratory, particularly Dr. Katelyn Walzer for ongoing discussions and feedback.

## Author Contributions

**Conceptualization:** Elizabeth D. English, Amandine Guérin, Boris Striepen.

**Formal analysis:** Elizabeth D. English, Boris Striepen.

**Funding acquisition:** Boris Striepen.

**Investigation:** Elizabeth D. English, Amandine Guérin, Jayesh Tandel.

**Resources:** Jayesh Tandel.

**Writing – original draft:** Elizabeth D. English, Amandine Guérin, Boris Striepen.

**Writing – review & editing:** Elizabeth D. English, Amandine Guérin, Boris Striepen.

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
