## [Editor Report · Decision Letter 0]

17 Dec 2021

Dear Dr. Striepen, 

Thank you for submitting your manuscript entitled "Live imaging of the Cryptosporidium parvum lifecycle reveals direct development of male and female gametes from type one meronts" for consideration as a Research Article by PLOS Biology.

Your manuscript has now been evaluated by the PLOS Biology editorial staff, as well as by an academic editor with relevant expertise, and I am writing to let you know that we would like to send your submission out for external peer review.

Once your full submission is complete, your paper will undergo a series of checks in preparation for peer review. Once your manuscript has passed the checks it will be sent out for review. To provide the metadata for your submission, please Login to Editorial Manager (https://www.editorialmanager.com/pbiology) within two working days, i.e. by Dec 19 2021 11:59PM.

If your manuscript has been previously reviewed at another journal, PLOS Biology is willing to work with those reviews in order to avoid re-starting the process. Submission of the previous reviews is entirely optional and our ability to use them effectively will depend on the willingness of the previous journal to confirm the content of the reports and share the reviewer identities. Please note that we reserve the right to invite additional reviewers if we consider that additional/independent reviewers are needed, although we aim to avoid this as far as possible. In our experience, working with previous reviews does save time. 

If you would like to send previous reviewer reports to us, please email me at pjaureguionieva@plos.org to let me know, including the name of the previous journal and the manuscript ID the study was given, as well as attaching a point-by-point response to reviewers that details how you have or plan to address the reviewers' concerns. 

Given the disruptions resulting from the ongoing COVID-19 pandemic, please expect some delays in the editorial process. We apologise in advance for any inconvenience caused and will do our best to minimize impact as far as possible.

Kind regards,

Paula

Paula Jauregui, PhD

Editor

PLOS Biology

---

## [Decision Letter · Decision Letter 1]

24 Feb 2022

Dear Dr Striepen,

Thank you for submitting your manuscript "Live imaging of the Cryptosporidium parvum lifecycle reveals direct development of male and female gametes from type one meronts" for consideration as a Research Article at PLOS Biology. Your manuscript has been evaluated by the PLOS Biology editors, an Academic Editor with relevant expertise, and by several independent reviewers.

You’ll see that all reviewers are quite excited about the work and they only raise minor points. Based on the reviews (pasted below), we will probably accept this manuscript for publication, provided you satisfactorily address the remaining points raised by the reviewers. Please also make sure to address the following data and other policy-related requests. Please note that we cannot make any decision about publication until we have seen the revised manuscript, as well as your response to the reviewers' reports and to our editorial requests. Your revised manuscript will be sent for further evaluation by the Academic Editor.

**Data and other policy-related requests:

1) Data: You may be aware of the PLOS Data Policy, which requires that all data be made available without restriction: http://journals.plos.org/plosbiology/s/data-availability. For more information, please also see this editorial: http://dx.doi.org/10.1371/journal.pbio.1001797

Note that we do not require all raw data. Rather, we ask for all individual quantitative observations that underlie the data summarized in the figures and results of your paper. For an example see here: http://www.plosbiology.org/article/info%3Adoi%2F10.1371%2Fjournal.pbio.1001908#s5

These data can be made available in one of the following forms:

I) Supplementary files (e.g., excel). Please ensure that all data files are uploaded as 'Supporting Information' and are invariably referred to (in the manuscript, figure legends, and the Description field when uploading your files) using the following format verbatim: S1 Data, S2 Data, etc. Multiple panels of a single or even several figures can be included as multiple sheets in one excel file that is saved using exactly the following convention: S1_Data.xlsx (using an underscore).

II) Deposition in a publicly available repository. Please also provide the accession code or a reviewer link so that we may view your data before publication.

Regardless of the method selected, please ensure that you provide the individual numerical values that underlie the summary data displayed in the following figure panels: Figures 1 CDF, 2 CDFGJK, 3 ABCDE, 4 C, 5 CGHI, Supplementary Figure 2C.

1.1) Please also ensure that each figure legend in your manuscript includes information on where the underlying data can be found and that your supplemental data file/s has/have a legend.

1.2) Please ensure that your Data Statement in the submission system accurately describes where your data can be found.

2) Blurb: Please provide a blurb which (if accepted) will be included in our weekly and monthly Electronic Table of Contents, sent out to readers of PLOS Biology, and may be used to promote your article in social media. The blurb should be about 30-40 words long and is subject to editorial changes. It should, without exaggeration, entice people to read your manuscript. It should not be redundant with the title and should not contain acronyms or abbreviations. For examples, view our author guidelines: https://journals.plos.org/plosbiology/s/revising-your-manuscript#loc-blurb

We expect to receive your revised manuscript within 1 month.

Please email us (plosbiology@plos.org) if you have any questions or concerns, or would like to request an extension.

**IMPORTANT - SUBMITTING YOUR REVISION**

*Resubmission Checklist*

*Published Peer Review*

Sincerely,

Dario

Dario Ummarino, PhD

Senior Editor

PLOS Biology

dummarino@plos.org

REVIEWS:

Reviewer #1: This a very exciting paper showing the specific timing and stages of the Cryptosporidium lifecycle. This paper is an incredible contribution to the field. The experiments are well thought out and clearly answer questions that have remained open in the field for a long time. The inclusion of the in vivo work really solidified the conclusions from he in vitro studies.

The only part of the paper that needs revision is the discussion, which, in stark contrast to the rest of the paper, is not well written. There are many run on sentences, sentences that are grammatically incorrect or simply unclear. "This" is used frequently without a noun following, which adds to the lack of clarity. There are missing commas, singular instead of plural words used (oocyst instead of oocysts) Additionally the discussion runs for 5.5 pages; it should be condensed considerably.

Reviewer #2: This is an important study that makes a profound advance to our understanding of the biology of this significant parasite. I teach a third year parasitology course- this study will cause me to change my lecture slides on the cryptosporidium lifecycle, and will necessitate a change to many new versions of textbooks on parasitology. The study reminds me of a quote by the old baseball catcher Yogi Berra who said "You can observe a lot by just watching".

Through their careful experiments, the authors establish that

*sexual differentiation appears to be triggered by a time/generation number rather than other external stimuli

*unlike the apparent case in other Apicomplexa, individual sexual stage producing parasites give rise directly to male and female gametes

*there is no compelling evidence for tetraploid type II meronts

The experiments are appropriately conducted and controlled, the experiments support the conclusions drawn, and the data are discussed fittingly and clearly. It is a pleasure to read.

Some suggestions:

1) The language in the abstract "We determine that the parasite executes an intrinsic program of three generations of asexual replication, followed by a single generation of sexual stages" is somewhat stronger than the "we propose that". The data for this experimental setup are very strong, but they don't exclude the possibility of other programs in other environmental conditions (including other in vivo scenarios), and I would encourage the use of some qualification of this statement in the abstract. Does this apply to all species? Is it true in other culture systems?

2) Figure 1

I didn't find 1A very clear - presumably this is meant to convey the two possibilities that commitment to sexual stages could be either through an external induction or intrinsic or time based pathway. But the arrows confused me - I initially processed this as a vertical arrow (pathway) coming down from "Extrinsic or intrinsic lifecycle progression" to the first green circle/parasite, and then found the other arrows confusing. Perhaps the source of confusion is that identically styled arrows in this diagram indicate transition from one parasite life stage to another as well as indicating external stimuli. Maybe it could be made clearer that there are two hypothesised pathways to commitment, with alternate arrows running through each pathway, while the external stimuli are symbolised using another symbol (lightning bolt? Or a different type of arrow)

3) I found the discussion starting at line 405 for the origin of the type II meront useful and interesting. However, I would appreciate a bit more detail here - the inference in this discussion is that the concept is an error based only on inference from the related Eimeria, but is there any other evidence for or against the existence of type II meronts? Is there other literature that tried to find these, or just repeated re-citation of the original Vetterling speculation that became inappropriately more dogmatic with time?

Minor Issues

4) I am not a Latin scholar, but the possessive s added to the end of a species or genus name (e.g. "Cryptosporidium's" at line 25 seems wrong to me. I would reword this to avoid that usage. I don't think a possessive S can be added to a formal scientific name, and several (admittedly unauthoritative ) style guides agree e.g. https://animaldiversity.org/teach/editor_guidelines/

https://markscherz.tumblr.com/post/85644377348/plurals-and-possessives-in-taxonomic-nomenclature

5) Figure 3

Although clearly described in the figure legend, some titles (beyond individual letters) above each panel would be helpful to avoid having to go back and forth between legend and graphic - e.g. "asexual" above panel B, "males" above C? 

6) I don't think the usage of "sexualises" in line 78 or "sexualized" at line 154 is correct. These normally mean to attribute a sexual role, but in these cases the meaning is that the parasites have been committed to a sexual lineage (or includes some parasites that are committed). I recommend rewording.

7) The sentence at line 97 reads strangely "The Cryptosporidium lifecycle is remarkably short and much of it unfolds in tissue culture." This gives the partial sense that the lab tissue flask is the natural habitat of the parasite. Do you mean that much of the lifecycle can be recapitulated in vitro/ ex vivo?

8) Line 183 "decerned". Decern is either archaic, or more recently has a judicial meaning that is not intended here. Discern is the relevant homophone.

9) I didn't understand the bit at lines 308-310 

"Based on our observations, the 95% confidence interval suggested that out of the 8 progeny in each meront, between 1.92 and 3.11 will be male. This is consistent with the fact that we never observed more than 2 male progeny from a single meront." If the confidence interval suggest that 3 of the 8 meronts can be male, why is this consistent with never seeing more than 2 males? 

10) Line 139 "We also conducted experiments using a C. parvum reporter strain [21]" 

I think this needs a bit more explanation without the reader having to go to this paper - e.g. that the strain expresses markers tagging male (Hap2?) and female (cowp1?) specific genes that allows the observer to discern male vs female gametes.

11) line 464 contrast should be contrasts

12) line 500 - see Bancells Nat Microbiol . 2019 Jan;4(1):144-154 for a potential exception (for mixed asexual and sexual, which would infer potential for later mix of male and female, though not directly from the same parasite as seen in this study

13) Figure 5 - Perhaps the figure would benefit from some arrows to more explicitly depict the sequence, as well as some annotation to clarify that the oocyst is a source of ongoing autoinfection as well as an infectious form that exits the host - I think this could be important for the many readers who aren't already familiar with the lifecycle.

a

Reviewer #3 (Guan Zhu): This manuscript presents solid and exciting data that corrects some long-term (mis)conception of some fundamental biology (i.e., the life cycle progression) on Cryptosporidium parvum - an important and globally distributed zoonotic apicomplexan parasite. 

It was a generally accepted/described (mis)conception that the life cycle of Cryptosporidium parasites undergoes 2-3 or more cycles of merogony (asexual development) to produce type I (8 nuclei) and type II (4 nuclei) meronts that produce 8 or 4 merozoites at the end of the merogonic cell cycles, but only merozoites produced by type II meronts enter gametogenesis (sexual development) to produce microgametes (male) and macrogametes (female). Using live imaging to track individual parasite cell development and progression, the authors provide solid data to clarify that: 

1) The parasite only produces 8-nuclear merozoites per asexual cell cycles, and the 4-nuclear meronts are mostly (if not all) transit form in the development from a 1-nuclear trophozoite to 8-nuclear meronts.

2) The merozoites produced after the third cycle of merogony could enter sexual development to form micro- or macrogametes. The general concept that sexual development starts from merozoites produced from type II meronts was incorrect.

3) The 8 individual merozoites produced from a single meronts may develop into either male or female gametes, rather than all or none to male or female. 

4) There are no external factors that initialize/trigger the sexual development. The sexual-ready merozoites are intrinsically and individually programmed for the fate to enter male or female development in the coming cell cycle. 

The findings are of high impact on the biology field in general, and on the parasitology field in particular. I have no major concerns on the data and conclusions, but a few minor comments as described below.

1) Figure 1, E-F (Super-infection assay data): Infection at 0 h = infection with H2bNeon strain of C. parvum (Cp) only; infection at 24 h = Mixed infections with both WT Cp (0 h) and H2bNeon strain Cp (infection at 24 h post-infection with WT Cp. It may be better to have a timeline bar (or may be another way) to clarify the two different types of h/hours in the figures.

2) Line 178 "found a mean time to egress of 12.57 hours": from which time point (e.g., from the recording of images at 11 hpi after inoculation of sporozoites, total = 11 + 12.57 = 23.57 hpi)?

3) Figure 2 (time points in lapse images and bar charts): In each panel, please clarify/calibrate the start point of the time in relationship to the h post-infection (hpi). Based on the description in lines 167-169 "We began imaging at 11 hours of cultures … also … imaging from 29 or 40 hours of culture onwards", the 0 h or 0:00 h start point could be equivalent to 11, 29 or 40 hpi.

4) Lines 208-218 (microgamete development): Please clarify the number of observed cell cycles for male parasites (4 cycles of cell division?).

5) Figure 3A, Pink/red lines: Pink lines are not well shown. Will another color work better, e.g., orange, plus some use of opacity? 

6) Figure 4: Please clarify the time points in each panel (equal to hpi or equal to +11 hpi). Figure 4B: Please clarify the meaning of the daggers. Figure 4C: Please clarify the "grey" zones (non-egressed merozoites?). Do all tracked meronts produced 8 merozoites?

7) Please define the "hours" in the manuscript when necessary (relationship with h post-infection [hpi]). Also most the words "hours" may be changed to "h".

8) Figure 6A: It may help to add arrows to between developmental stages, and mark both male and female gamonts and gametes. In current form, it appears that the female gametes are developed from male gamonts.

---

## [Editor Report · Decision Letter 2]

11 Mar 2022

Dear Dr Striepen,

On behalf of my colleagues and the Academic Editor, Gary Ward, I am pleased to say that we can in principle accept your Research Article "Live imaging of the Cryptosporidium parvum lifecycle reveals direct development of male and female gametes from type one meronts" for publication in PLOS Biology, provided you address any remaining formatting and reporting issues. These will be detailed in an email that will follow this letter and that you will usually receive within 2-3 business days, during which time no action is required from you. Please note that we will not be able to formally accept your manuscript and schedule it for publication until you have any requested changes.

PRESS

Sincerely, 

Dario

Dario Ummarino, PhD 

Senior Editor 

PLOS Biology

dummarino@plos.org